# Unveiling the Role of Dissolved Organic Matter on the Hg Phytoavailability in Biochar-Amended Soils

**DOI:** 10.3390/ijerph20043761

**Published:** 2023-02-20

**Authors:** Wenhao Chen, Zhigang Yu, Xu Yang, Tantan Wang, Zihao Li, Xin Wen, Yubo He, Chang Zhang

**Affiliations:** 1College of Environmental Science and Engineering, Hunan University, Changsha 410082, China; 2Key Laboratory of Environmental Biology and Pollution Control (Hunan University), Ministry of Education, Changsha 410082, China; 3Australian Centre for Water and Environmental Biotechnology (Formerly AWMC), The University of Queensland, Brisbane, QLD 4072, Australia

**Keywords:** biochar, mercury, pyrolysis temperature, dissolved organic matter, soil remediation

## Abstract

Biochar can effectively reduce the phytoavailability of mercury (Hg) in soil, but the mechanisms are not fully understood. In this study, the dynamic changes in Hg content adsorbed by the biochar (BC-Hg), Hg phytoavailability in the soil (P-Hg), and soil dissolved organic matter (DOM) characteristics were determined over a 60-day treatment period. Biochar obtained at 300 °C, 500 °C and 700 °C reduced the P-Hg concentration assessed by MgCl_2_ extraction by 9.4%, 23.5% and 32.7%, respectively. However, biochar showed a very limited adsorption on Hg, with the maximum BC-Hg content only accounting for 1.1% of the total amount. High-resolution scanning electron microscopy coupled with energy dispersive X-ray spectroscopy (SEM-EDS) results showed that the proportion of Hg atoms in biochar after 60 d was barely detectable. Biochar treatment can shift soil DOM toward higher aromatic content and molecular weight. Additionally, the addition of high-temperature biochar increased more humus-like components, but low-temperature biochar increased more protein-like components. Correlation analysis and partial least squares path modeling (PLS-PM) showed that biochar promoted humus-like fractions formation to reduce the Hg phytoavailability. This research has deepened the understanding of the mechanisms by which biochar stabilizes Hg in agricultural soils.

## 1. Introduction

Heavy metal pollution is a growing environmental problem worldwide and has attracted widespread attention [1,2]. Mercury (Hg), an extremely toxic element, can damage the nervous system of animals and humans. In addition, Hg tends to accumulate and amplify via the food chain, eventually constituting a health risk and carcinogenicity to humans, and even lives in the environment exposed to low concentrations of Hg [3]. In order to assess the impact of Hg on soil organisms, there is a growing tendency to use the concept of availability [4,5]. Availability can determine the fate of metals in the soil and their ability to be leached into groundwater or taken up by plants. To this end, it is required to find ways to reduce Hg availability to plants by retaining it in the soil or converting it in situ into its most stable and less toxic form [6,7].

Biochar is a sustainable carbon-rich material made by the cracking of biomass in an oxygen-limited environment [8]. Owing to its high adsorption capacity for heavy metals, biochar is often used for soil remediation [9,10]. Biochar has shown excellent performance in Hg immobilization, but it has been less researched, other than the metal(loid)s such as Pb, As, and Cd. In particular, the pathway of Hg immobilization by biochar is still unclear. On the one hand, the addition of biochar can adsorb heavy metals to reduce availability [11,12,13]. On the other hand, biochar inevitably leads to a series of biogeochemical interactions in the soil matrix, causing changes in soil properties (e.g., soil pH and DOM properties), which affect the immobilization of heavy metals [14,15,16]. Biochar changes soil DOM both by releasing DOM (i.e., the easily mineralizable and extractable carbon fraction in biochar) into the soil solution and adsorbing the native soil DOM into its pore structures [17]. Given the high affinity of Hg for DOM, the effect of biochar-influenced DOM on Hg phytoavailability should be carefully considered.

The mobility and availability of heavy metals are affected by its capacity to form soluble or insoluble complexes with soil DOM and its binding properties to DOM, which are mainly associated with the structure and composition of DOM [18,19]. A number of studies have investigated the influence of various chemical functional groups, compositional structures, and sources of DOM on Hg availability [20,21,22]. For example, highly aromatic humic-like substances can be amended to Hg-contaminated soil and it shows a higher Hg binding capacity [23]. Combined with the results of existing studies, it has been initially discovered that biochar appears to affect soil DOM properties such as the components and content, and thus affects the Hg availability [24,25]. The pyrolysis temperature was one of the key factors that significantly influenced the content, structure and composition of biochar-derived DOM [26,27]. However, little is known about the changes in the composition of soil DOM and its corresponding effects on Hg phytoavailability when applying biochar produced at different pyrolysis temperatures.

In this study, we hypothesized that biochar-induced variations in soil DOM properties and therefore Hg–soil interactions play key roles in controlling the Hg phytoavailability. To test this hypothesis, we designed a nylon bag-biochar incubation experiment. Specifically, a certain amount of rice straw biochar with three different pyrolysis temperatures (300 °C, 500 °C, 700 °C) was placed in a nylon mesh bag and then added to the Hg-contaminated soil. The main objectives of this study were (1) to clarify the main action mode of biochar on Hg by assessing the adsorbed Hg content by biochar and the available Hg content in soil; (2) to determine the effect of biochar produced at different pyrolysis temperatures on the soil DOM properties; and (3) to reveal the potential mechanisms by which biochar immobilizes Hg in soil. The findings from this study could assist in improving the mechanistic understanding of Hg availability in soils under biochar amendment as well as provide a theoretical guidance for the practical application management of biochar.

## 2. Material and Methods

### 2.1. Preparation of Biochar and Soil

The soil samples used in this study were collected from farmland in Yongzhou, Hunan Province. Soils with a depth of 0–10 cm were used in the experiments, which were air-dried, mixed, sieved to a diameter of ≤2 mm and stored in a greenhouse (~22 °C) prior to the experiments. The basic characteristics of the soil are shown in Appendix A and its operation is described in the Appendix A.

Since Hg concentrations in various human-influenced contaminated soils vary in the range from 0.04 to >100 mg kg^−1^ [20,28], the Hg concentrations chosen in previous experimental studies are not uniform. We chose a soil Hg concentration of 20 mg kg^−1^ because this is representative of the typical Hg-contaminated agricultural soils in mining areas [29]. Specifically, a calculated solution of Hg(NO_3_)_2_ was added uniformly to 1 kg of air-dried soil, mixed continuously and allowed to increase its moisture content to 20% *w*/*w* [30]. The soil was then covered with dark plastic lids (to reduce the potential losses of Hg) and incubated for 7 days to homogenize the soil [31].

Biochar was made from rice straw. The straw was heated in a tubular furnace with limited air at a rate of 10 °C min^−1^. In the temperature range (200–700 °C) applied in the previous studies [30,32], the pyrolysis temperatures 300, 500 and 700 °C were used. After reaching the final pyrolysis temperature, the pyrolysis residence time was 120 min. The choice of pyrolysis temperature was determined by the particular physicochemical changes that each temperature can produce for the biochar. Increasing the pyrolysis temperature (~700 °C) is beneficial to both increase the specific surface area (SSA) and ash content of the biochar [33]. However, it reduces the dissolved organic carbon (DOC) content. In contrast, biochars pyrolyzed at lower temperature (~300 °C) have more functional organic groups, more DOC content and lower SSA [34]. The choice of a pyrolysis temperature of 500 °C was based on the possibility of exhibiting the intermediate properties at both low- and high-temperature biochars. Changes in these properties by pyrolysis temperature may have an impact on the biochar’s ability to adsorb and release DOM, which in turn determine the effectiveness of Hg immobilization. The basic characteristics of biochar were analyzed and its specific operations are shown in the Appendix A.

### 2.2. Experiment Design

Each biochar was sieved to obtain the particle sizes of 0.15 to 0.5 mm, which is often found in biochar-treated Hg-contaminated soils [30]. The selected application rate of biochar was 5%, and this application rate was widely used in the studies of Hg-contaminated soil [35,36]. The biochar was packed into acid-washed microporous mesh nylon bags (pore size 0.05 mm). This was aimed at preventing biochar particles from leaching into the soil, as well as allowing the transport of Hg [37,38]. The bagged biochar and 12 g of soil were placed sequentially in a 50 mL polypropylene centrifuge tube so that the biochar could be uniformly and tightly encapsulated by the soil.

This experiment was set up with four treatment groups: control (CK), 300 °C biochar treatment (3BC), 500 °C biochar treatment (5BC) and 700 °C biochar treatment (7BC). Each group had three replicates. All the treatments were incubated in a constant temperature chamber at 28 °C. Soil moisture content was controlled at 70% of the water holding capacity (WHC). The moisture content was adjusted every two days by adding deionized water to maintain a constant mass. Samples were destructively sampled on days 5, 15, 30, 45 and 60 of the incubation experiment.

### 2.3. Sampling and Chemical Analyses

During each destructive sampling, the biochar-treated soil and bagged biochar were collected separately from each treatment sample for further analysis. Soil pH was measured with a pH electrode using a solid/water ratio of 1:2.5 (*w*/*v*). The chemical functional groups of the soil were characterized using infrared spectroscopy (IRAffinity-1, Shimadzu, Kyoto, Japan). Before analysis, soil samples and potassium bromide were mixed and pressed at a ratio of 1:150 to prepare thin slices. The scanning wavelength was 4000–400 cm^−1^, with a resolution of 4 cm^−1^ and a scan number of 32. The infrared spectra were processed using the EZ OMNIC 7.3 software. The quantification of the Hg phytoavailability in biochar-treated soils and the total Hg in biochar was undertaken to determine the main way in which biochar acts to immobilize Hg in contaminated soils. The phytoavailable Hg was analyzed at a ratio of 1M MgCl_2_: soil of 1:10 with shaking at 200 rpm for 1 h. It was found that the MgCl_2_ extractable level of Hg was representative of the Hg phytoavailability and showed a good correlation with Hg content in the plant body [39,40,41]. The Hg content was then measured by cold vapor atomic fluorescence spectrometry (CVAFS) [42]. Scanning electron microscopy (SEM) with energy dispersive spectrometry (EDS) (ZEISS Gemini SEM 300, Oberkochen, Germany) was used to determine the morphology and chemical composition of the biochar particles, as described in the Appendix A.

### 2.4. Characterization of Water-Extractable DOM

The extraction of DOM followed the procedure previously described in a related study [43]. Prior to analysis, the filtrate was stored in the dark at 4 °C. The concentration of DOC in the extracted DOM was measured using a TOC analyzer (Shimadzu L series, TOC-CHP, Kyoto, Japan). The DOC of all the samples was uniformly adjusted to 10 mg L^−1^ to avoid the inner-filter effect prior to UV–Vis and fluorescence analysis [44]. The UV absorbance of DOM was measured using a 1-cm quartz cuvette and a UV/Vis spectrophotometer (Perkin-Elmer Lambda 14). The scanning was performed with deionized water as the blank and operated with scanning wavelengths of 200–600 nm. Detailed information about the procedures and parameters of parallel factor (PARAFAC) analyses and simultaneous fluorescence spectroscopy (SFS) can be found in the Appendix A. Several spectral parameters were used to characterize DOM, including SUVA_254_ (specific UV-absorbance values at 254 nm), E_2_/E_3_ (absorbance at 254 nm divided by absorbance at 365 nm), Frl (freshness index), FI (fluorescence index) and BIX (biological index) and HIX (humification index). Details of the calculation of the DOM spectral parameters are given in Appendix A.

### 2.5. Statistical Analysis

Tests of variance for the non-normally distributed data set were performed by Kruskal–Wallis one-way ANOVA. Statistical significance (*p*) was declared at <0.05 (two-tailed). Correlation analysis was based on Spearman’s rank correlation coefficient (r_s_) at the *p* < 0.01 or *p* < 0.05 level. All graphs were plotted by Origin 2022b. Partial least squares pathway modeling (PLS-PM) was performed using Smart-PLS 3 to assess the effect and relative contribution of biochar-induced soil pH, DOC and DOM characteristics (e.g., components and properties) on Hg phytoavailability. PLS-PM was first conducted to show the associations among all the variables (full model), and then we kept the variables with loadings greater than 0.4 to the latent variable for further PLS-PM analysis [45]. Finally, the DOM components include the relative abundance of the three DOM components. The DOM properties are the set of sources and molecular weights of DOM represented by BIX, Fl and E_2_/E_3_.

## 3. Results and Discussion

### 3.1. Effect of Biochar on Hg Phytoavailability in Soil

In order to reveal the potential mechanism of biochar-induced Hg phytoavailability in soil, we investigated the phytoavailable Hg content in the “biochar-treated soil” and the Hg content of biochar through adsorption.

The reduction of P-Hg in the soil by biochar was significant. The treatment groups of 5BC and 7BC resulted in a rapid reduction in phytoavailable Hg content in the soil within 5 d compared to CK (Figure 1e). Consistent with previous studies, this work observed that the treatment effect of biochar could be achieved in a relatively short time using biochar (550 °C) to immobilize 28.3 mg/kg Hg-contaminated soil [46]. The largest reductions of 3BC, 5BC and 7BC occurred at 30 days, which declined by 15%, 40% and 47%, respectively. This was followed by a slight increase in P-Hg concentrations. This work shows that rice straw biochar without modification can effectively immobilize Hg. In addition, the results revealed that the pyrolysis temperature of biochar made a significant difference to Hg phytoavailability.

The SEM-EDS results showed that the surface structure of biochar became more porous as the pyrolysis temperature increased, which was also supported by N_2_-BET results that biochar produced at the highest temperature had the largest specific area (Appendix A). The fresh biochar had a relatively smooth and porous surface with a high content of C and the presence of Mg, K, Ca, Fe and Mn. Biochar recovered after 60 d had visible microaggregates on the surface and in pores, with higher concentrations of Al, Fe, Mn and Ca than fresh biochar. However, Hg was below the detection limit in biochar (Figure 1d). Furthermore, we found that the adsorption of Hg in soil by 3BC, 5BC and 7BC (Figure 1f) was 1.434–2.491 mg/kg, 2.756–4.414 mg/kg and 2.405–4.113 mg/kg, respectively, by ablation-CVAFS analysis. The maximum amount of BC-Hg was only 1.1%, which indicated that the Hg adsorption by biochar was insignificant. While biochars have been often applied as an amendment to contaminated soils, there are few comparative sorption data. A similar study has shown that biochar exhibits low sorption capacity in situ treatment of Hg-contaminated sediments [47,48]. In fact, the effectiveness of in situ amendments to Hg is based on stoichiometric considerations (the amendments active binding sites should exceed that of Hg in sediments) and the distribution coefficients (Kd) of the amendment for Hg since the amendment functions by competing Hg against the natural sorbents [49]. In this study, the main reason for this phenomenon could be that Hg readily formed complexes with soil organic matter [31,50], reducing free Hg adsorption on biochar. Overall, our results indicate that biochar has limited adsorption to Hg in soil.

### 3.2. Changes in Physicochemical Properties of Soil

The results of soil infrared spectroscopy with different biochar amendments (Figure 2) showed that the chemical functional group composition of the soil with biochar and the soil without biochar was basically similar, but the peak intensities of these characteristic peaks were different. According to previous studies, the peak at 790 cm^−1^ was attributed to the C-H bending vibration of the benzene ring; the peak at 925 cm^−1^ was attributed to the basic vibration peak of illite; the peak at 1032 cm^−1^ was attributed to the C-O stretching of polysaccharides, alcohols and phenols vibration; and the 1677 cm^−1^ was attributed to the aromatic C=C stretching vibration [51,52]. Compared with CK, the absorption peak intensity of 5BC and 7BC at 1677 cm^−1^ increased slightly, and the absorption peak height of 3BC was basically similar to that of CK. On the contrary, the absorption peak intensity at 1032 cm^−1^ was slightly decreased for 5BC and 7BC, but increased significantly for 3BC. In addition, 3BC also significantly improved the absorption peak intensity at 790 cm^−1^. This result suggests that differences in the functional groups of soil organic matter are associated with the properties of biochar prepared at different pyrolysis temperatures.

With the addition of biochar, the soil pH has improved considerably in the short term (Appendix A). Compared to CK, the pH increased by 0.2, 0.5 and 0.65 after 60 d for 3BC, 5BC and 7BC, respectively (Table 1). This is mainly related to the high alkalinity of biochar, which contains large amounts of alkali ions that replace H^+^ in soils and thus improves pH (Figure 1d). All biochar treatments increased soil DOC content, and in particular, the biochar obtained at higher pyrolysis temperature increased the DOC content more significantly. The maximum soil DOC content was 913.17, 949.78 and 1073.68 mg/kg for 3BC, 5BC, and 7BC, respectively. However, the DOC content of the biochars declined markedly with increasing production temperature (Appendix A). This result suggests that the DOC release from biochar to the environment may be influenced by other factors. We found that pH was significantly and positively correlated with DOC content (r_s_ = 0.562, *p* < 0.01), which was in line with previous studies [14,19]. This may be attributed to the presence of C-O-C bonds in the cellulose and hemicellulose of biochar which broke rapidly with increasing pH [53]. After 30 d, DOC concentration in the biochar-treated group showed a significant decreasing trend over time, leading to a reduction in DOC differences between the treatment groups. This may be related to microbial depletion in the soil [54,55].

Soil DOC and pH are two key factors for biochar to mitigate the biotoxicity of heavy metals [15,56]. We observed a negative correlation of P-Hg with pH (r_s_ = −0.644, *p* < 0.01) and DOC (r_s_ = −0.692, *p* < 0.01), respectively. At higher pH (alkaline conditions), metals tend to produce more stable forms, while at lower pH (acidic conditions) they tend to occur as soluble organometallic or free ionic species with high availability [57,58]. In addition, an increase in soil pH can also cause an increase in the net negative charge of soil surfaces, promoting the adsorption of metals on the soil surface and reducing metal utilization [59]. The interaction between DOM released from biochar-amended soils and metals can effectively control the availability of Hg [25,60,61]. However, the opposite result was also observed in previous studies, showing that mobility and availability of Hg in soil increased with increasing DOC content [19,20]. This paradoxical phenomenon may be due to the diversity of DOM. Therefore, the simple correlation between DOC concentrations and phytoavailable Hg does not adequately reflect its intricate relationship.

### 3.3. Effect of Biochar under Different Pyrolysis Temperatures on the Spectral Properties of Soil DOM

#### 3.3.1. UV–Vis Absorption Spectral Characteristics

UV–Vis spectroscopy showed that the degree of shift in DOM characteristics in soils was different. Compared to the 5th day, the absorption rate at shorter wavelengths in each treatment group decreased significantly in the 60th day (Appendix A). SUVA_254_ has been widely used to evaluate the aromaticity of DOM [62]. The aromaticity and molecular weight of DOM have a positive relationship with SUVA_254_ and an inverse relationship with E_2_/E_3_ values [63], which was further confirmed by a negative correlation between SUVA_254_ and E_2_/E_3_ of DOM samples in this experiment (r_s_ = −0.818, *p* < 0.01). Previous studies showed that biochar amendments enhance the DOM released from soils as well as shift DOM composition toward higher aromatic content [14]. In our experiments, the aromaticity and molecular weight of soil DOM increased with increasing the applied biochar pyrolysis temperature in the other treatment groups, except for 3BC that slightly decreased the aromaticity and molecular weight of soil DOM (Figure 3). This is because the aromaticity and molecular weight of DOM released from biochar gradually increase with increasing pyrolysis temperature [64]. Additionally, the aromatic component of organic matter is difficult to be accessed by microorganisms due to its hydrophobicity and possible toxicity [65]. Therefore, high-temperature biochar DOM remained in the soil. The aromaticity and molecular weight of soil DOM increased in each treatment group with incubation time (Figure 3), which was in line with a former study [66]. This indicates a gradual release of aromatic substances or consumption of small non-aromatic substrates by microorganisms. Overall, the soil DOM properties were influenced by the biochar addition and varied with the pyrolysis temperature.

#### 3.3.2. Parallel Factor Analysis and Fluorescence Indicators

The different components of soil DOM play critical roles in regulating the fate of Hg [67]. Therefore, soil DOM fractions were investigated. A total of three components (C1–C3) were extracted from the soil samples (Figure 4). The excitation and emission characteristics of the components and its comparison with former studies are shown in Appendix A. Component C1 (250, 310/425 nm) was similar to the traditional combination of humic-like peaks A and M and was described as a fulvic acid-rich humic substance of terrestrial origin [68,69]. This component was generally considered to be a low molecular weight compound from terrestrial plants or soil organic matter related to biological activity [70,71]. Component C2 (267, 370/480 nm) was considered to be a humus-like component of terrestrial origin [72]. This excitation and emission property were related to terrestrial organic matter consisting of high molecular weight and aromatic organic compounds [73]. The results showed that SUVA_254_ was positively correlated with C2% (Appendix A), indicating that Component C2 consists of compounds with relatively high aromaticity and molecular weight. Component C3 (<240, 290/375) was derived from authigenic protein-like material associated with microbial degradation [68,74]. The relative abundance of C3 was negatively correlated with HIX (r_s_ = −0.583, *p* < 0.01) and positively correlated with recently produced DOM, suggesting that Component C3 consists of DOM produced by indigenous microorganisms. Based on the fluorescence spectra, the molecular structure of component C3 was relatively simple [75].

Among the three components analyzed by EEM, the fluorescence intensity of the fulvic acid-rich humus (C1) was the highest (Figure 4), indicating that the DOM components of the studied soil were dominated by fulvic acid-rich humus. Compared to CK, the application of biochar increased the relative abundance of the three compositions in the soils. However, the biochar produced at higher pyrolysis temperature contributed more humic-like fractions to the soil DOM. In addition, the degree of soil DOM humification (HIX, Appendix A) increased following high-temperature biochar application, which seemed to be influenced by the release of exogenous DOM from organic amendments [14]. Conversely, more protein-like fraction and higher autotrophic capacity (e.g., Frl, BIX, Fl, Appendix A) were observed in the soil DOM treated with low-temperature biochar. This indicated that the microbial activity in the 3BC soil was enhanced under stimulation of exogenous nutrients and energy. The changes in the composition of fluorophores seemed to complement and confirm the results of UV–Vis spectra. The high aromaticity and molecular weight, and the high humification degree of DOM in biochar-treated soil can be attributed to the increase of humic-like, aromatic and condensed aromatic components [76]. The lower SUVA_254_ values in the 3BC treatment (Figure 3) indicated that the 3BC treatment contained more soluble substances in the soil that were easily absorbed by soil microorganisms [77]. Therefore, the autochthonous contribution in the soil DOM of the 3BC treatment was increased. Overall, compared with low-temperature biochar inputs, high-temperature biochar could promote more aromatic and large molecular weight humus into soil and the concomitant low bioavailability for soil microorganism. These differences may influence the binding capacity of DOM to Hg, as spectral characteristics are important factors controlling DOM complexation with metals [78,79].

#### 3.3.3. 2D-COS Analysis of Fluorescent Components

To illustrate the effect of biochar on the degree of compositional variation of soil-derived DOM, 2D-COS analysis was performed (Figure 5). In the synchronous plot, the auto-peaks attributed to protein-like substances at 267 and 294 nm and the auto-peaks attributed to fulvic acid-like substances at 330 and 370 nm were mainly observed to be concentrated [80]. In the CK group, the intensity of the auto-peaks decreased in the order of 330 > 267, 370 > 294 nm; in 3BC, the intensity of the self-peaks decreased in the order of 330 > 294, 370 > 264 nm; in 5BC and 7BC, the intensity of the self-peaks decreased in the order of 330 > 370 > 294 > 267 nm. This indicated that the fluorescence of the fulvic acid-like components of the soil was more susceptible than that of the protein-like material. In the four synchronous maps, the cross-peaks were positive except at 267 nm, indicating that the peak intensities at 294 nm and 330, 370 nm showed the same variation trend, while the peak at 267 nm showed the opposite trend with the incubation process.

In the asynchronous map of the CK group, five cross-peaks located at the lower right corner were observed, and according to Noida’s rule [81], the order of wavelength change with time was: 267→330→370, 420 nm; 267→294 nm. In the 3BC and 5BC groups, four cross-peaks were observed in the asynchronous map, and the order of wavelength changes with time was: 267→330→370 nm; 294→370 nm and 267, 294→370 nm; 330→370 nm, respectively. The degree of variation in the fractions of the CK, 3BC and 5BC treatment groups were similar, and the all results indicated that the protein-like substances changed more strongly than the fulvic acid-like substances. In the asynchronous map of the 7BC group, five cross-peaks were observed in the lower right corner, and the wavelengths changed with time in the following pattern: 330→267, 380→405 nm; 330→360 nm. The 7BC group showed stronger changes in fulvic-like substances than humic-like and protein-like substances. The different changes presented by the biochar treatments illustrated the effect of pyrolysis temperature on the changes in soil DOM composition. In addition, the variation of wavelength with time was different in the same fluorescent component, showing the heterogeneity of DOM.

### 3.4. Correlation of DOM Properties with Phytoavailable Hg

The Spearman’s rank correlations between DOM properties and Hg phytoavailability in soil are summarized in Appendix A. Biochar treatments can lead to higher aromaticity, molecular weight of soil DOM (Figure 3). This facilitates organics–metal binding, followed by the formation of high molecular weight organic matter–metal complexes and/or attachment to the surface of soil particles, thereby reducing the solubility and availability of metals [21,82]. In general, hydrophobic fractions of DOM, particularly humic molecules, have a high content of reduced sulfur (e.g., sulfides and thiols), which causes a preferential affinity for Hg [83,84]. There was a negative correlation between P-Hg and the relative abundance of C1 (r_s_ = −0.489, *p* < 0.05) and C2 (r_s_ = −0.495, *p* < 0.05), but no significant relationship with the relative abundance of C3 (*p* > 0.05), implying that less humified organic matter has a lower capacity and strength to bind Hg than recalcitrant organic matter [21]. However, a significant negative correlation of P-Hg values with C2% was observed in our experiments, but not in C1%. This may be due to the instability of the C1 component. In most cases, the stability of metal–organic complexes increased with increasing soil pH, with higher stability of humic-like substances compared to fulvic-like substances (acidic soil pH) [85,86]. This was further backed up by the negative correlation between C1% and pH (r_s_ = −0.439, *p* < 0.05) and the positive correlation between C2% and pH (r_s_ = 0.919, *p* < 0.01) in the studied soils. In the contaminated soils, the decomposition of the C1 component may led to the release of Hg originally bound to organic matter, which helps to understand why P-Hg values increase slowly in the later stages of incubation. In addition, the P-Hg was significantly and positively correlated with the C3% (r_s_ = 0.589, *p* < 0.01). Consistently, Frl was positively correlated with P-Hg (r_s_ = 0.529, *p* < 0.05). This suggests that the microbial source of fresh DOM seems to bring a negative effect on Hg immobilization. The high E_2_/E_3_ and BIX values in 3BC may imply a significant DOM contribution from low molecular weight microbial sources, while the opposite is true for high-temperature biochar. This explains the differences in Hg immobilization by biochar that was produced at different temperatures. Overall, humus-like components with higher aromaticity and molecular weight are difficult to be biodegraded and thus present relatively high complexation ability to reduce the Hg phytoavailability [26]. Protein-like substances with relatively simple and unstable molecular structures may release “new” Hg from the soil through mineralization of organic matter and microbial metabolism [31,87].

### 3.5. PLS-PM Analysis

Considering the limited number of samples, further multivariate statistical analysis by PLS-PM was chosen to assess the effect of biochar-induced changes in soil properties on Hg phytoavailability (Figure 6). The good predictive value of the model was demonstrated by the goodness-of-fit (GOF) of 0.733. The direct effect of DOC content on P-Hg was the largest (−0.467). In contrast, the DOM components had a significant negative effect on P-Hg (−0.258), although its effect was less than that of DOC. This indicates that biochar-induced changes in DOM fractions also play an important role in reducing the phytoavailable Hg. Interestingly, our PLS-PM showed that the direct effect of pH on P-Hg was not significant. This means that pH may indirectly control Hg mainly by promoting changes in soil DOM concentration and fraction. Based on this work, biochars produced at low temperatures may not be suitable for Hg immobilization because of its low improvement in the soil properties. Biochars produced at 500 °C or higher may be more appropriate for practical applications of Hg immobilization.

### 3.6. Cost-Benefit Analysis of Using Biochar

The economic costs of biochar remediation often determine whether it is feasible in practice [88]. The amount of utilized biochar for Hg-contaminated soil remediation is shown in the Appendix A (about 100 tons/ha). Assuming that the average price of biochar is about USD 300/ton [25], the rough cost of biochar remediation of contaminated farmland at this application rate is USD 30,000/ha. The economic cost is lower than the average of existing technologies for the remediation of heavy metal-contaminated soil (about 40,000 USD/ha) in China [89]. Therefore, it may be economically acceptable to use this application rate. Moreover, biochar may increase crop yields while immobilizing Hg [90,91]. A previous study carried out a meta-analysis of the relationship between biochar and crop yields, and showed that the overall positive effect of biochar on crop yields was about 10% with high statistical significance [92]. The greatest positive effect was seen in the biochar application rate of 100 t/ha (approximately 5% wt., Appendix A). Therefore, when viewed in combination with the environmental benefits from metal immobilization and the yield benefits from plant promotion, this biochar application rate may provide a viable option for maximizing economic benefits.

## 4. Conclusions

This study showed that biochar reduced the Hg phytoavailability mainly by altering soil DOM properties rather than by its sorption. The application of biochar improved soil pH and increased the DOM composition with aromatic and high molecular weight. In particular, the addition of high-temperature biochar significantly increased the humic-like components of the soil. Humus-like components were effective in reducing Hg phytoavailability by forming stable complexes with Hg. Altogether, these results suggest that biochar (especially the high-temperature biochar) increases the highly aromatic humic-like substances to immobilize Hg in contaminated soils. In fact, the factors affecting Hg availability in the rhizosphere soil environment are extremely different from those in the non-rhizosphere soil environment. Therefore, future work needs to focus on the impact of soil constituents (e.g., root exudates, minerals and microorganisms) on the Hg immobilization by biochar in long-term field experiments.

## Figures and Tables

**Figure 1 ijerph-20-03761-f001:**
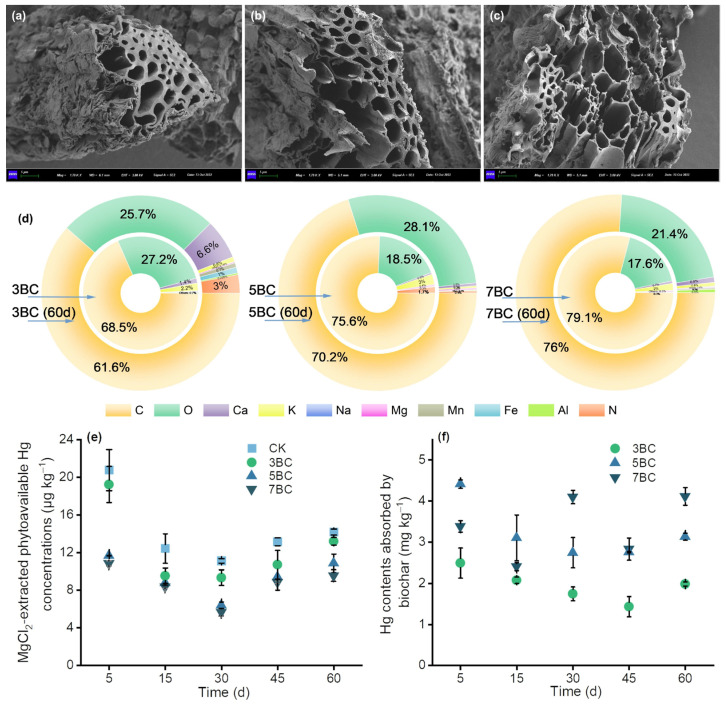
SEM images of fresh biochar at 300 °C (**a**), 500 °C (**b**) and 700 °C (**c**). The element composition of the biochar removed from soil (**d**). The phytoavailable Hg contents in the soil (**e**) and the concentration of Hg in BC (**f**). Data are presented as mean ± standard deviation (SD), n = 3. CK, control; 3BC, 300 °C biochar treatment; 5BC, 500 °C biochar treatment; 7BC, 700 °C biochar treatment.

**Figure 2 ijerph-20-03761-f002:**
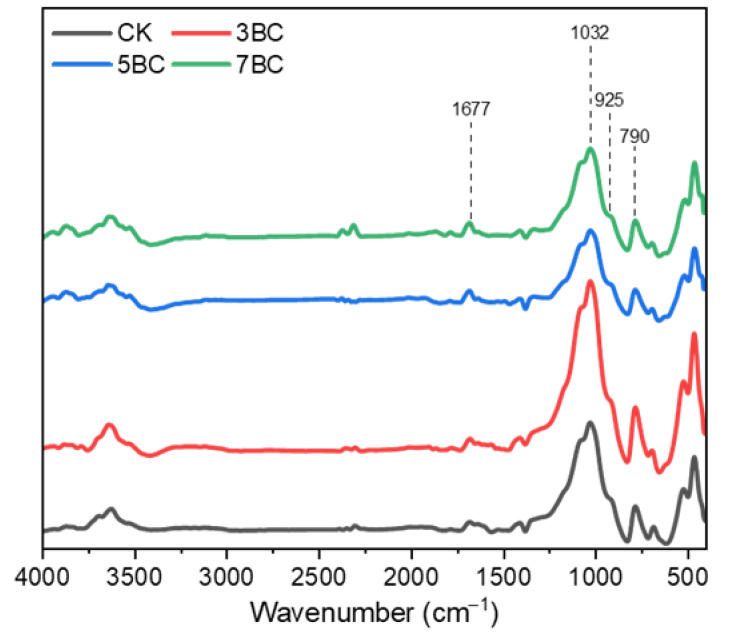
Infrared spectra of soil after 60 d of incubation. CK, no biochar amendment; 3BC, 300 °C straw biochar amendment; 5BC, 500 °C straw biochar amendment; 7BC, 700 °C straw biochar amendment.

**Figure 3 ijerph-20-03761-f003:**
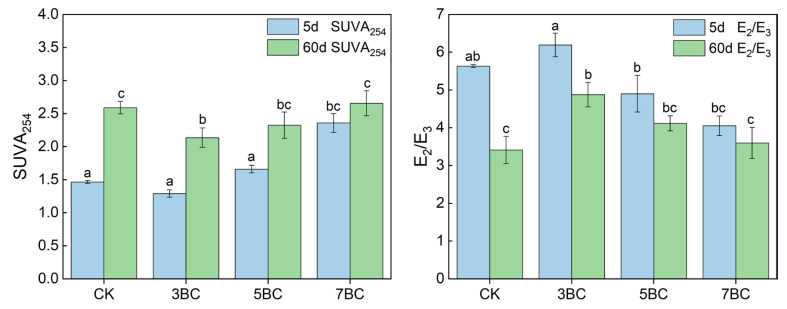
SUVA_254_ and E_2_/E_3_ of DOM in soil treated with biochar at different incubation times (5 d and 60 d). Data are presented as mean ± standard deviation (SD), n = 3. Different letters indicate significant differences between treatments of the soil (one-way ANOVA, *p* < 0.05), respectively. CK, control; 3BC, 300 °C biochar treatment; 5BC, 500 °C biochar treatment; 7BC, 700 °C biochar treatment.

**Figure 4 ijerph-20-03761-f004:**
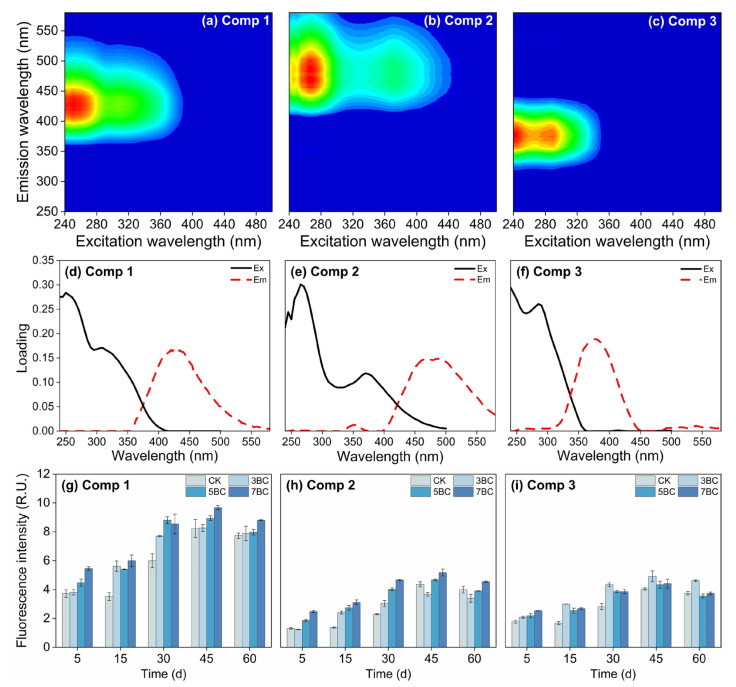
Three fluorescent components (C1: humic-like component enriched with fulvic acids, (**a**); C2: humic-like component, (**b**); C3: protein-like component, (**c**)) and corresponding excitation/emission loads (**d**–**f**) in biochar-treated soils as determined by EEM-PARAFAC analysis, and the fluorescence intensity (**g**–**i**) of the components during incubation. Data are presented as mean ± standard deviation (SD), n = 3. CK, control; 3BC, 300 °C biochar treatment; 5BC, 500 °C biochar treatment; 7BC, 700 °C biochar treatment.

**Figure 5 ijerph-20-03761-f005:**
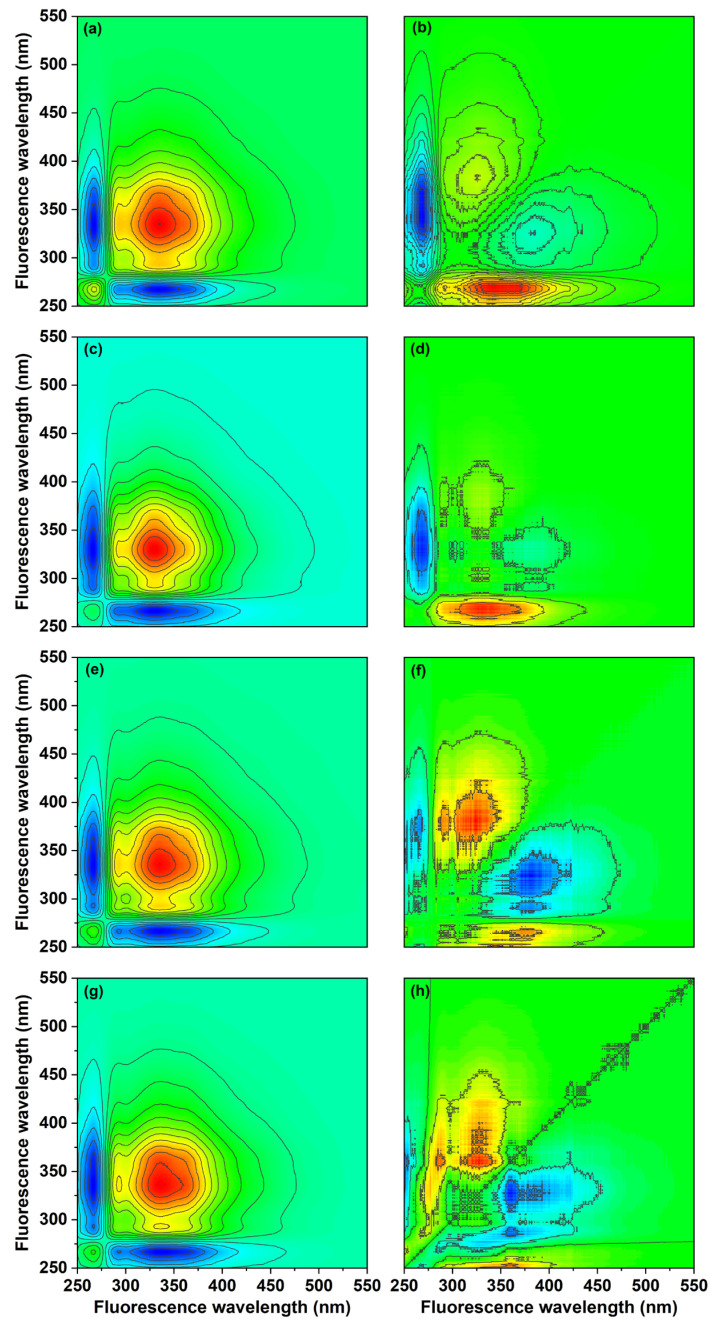
2D-COS maps generated by synchronous fluorescence spectra of soil DOM in the 250–550 nm region. Synchronous map (CK: (**a**); 3BC: (**c**); 5BC: (**e**); 7BC: (**g**)) and asynchronous map (CK: (**b**); 3BC: (**d**); 5BC: (**f**); 7BC: (**h**)) for different treatment groups. CK, control; 3BC, 300 °C biochar treatment; 5BC, 500 °C biochar treatment; 7BC, 700 °C biochar treatment.

**Figure 6 ijerph-20-03761-f006:**
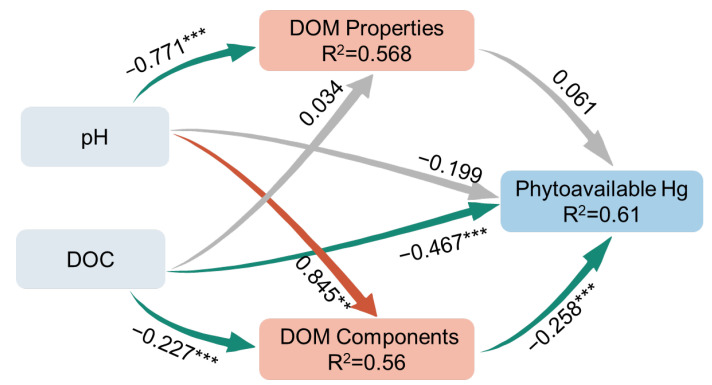
PLS-PM analysis of relationships between soil pH and DOC, DOM characteristics (components and properties), and phytoavailable Hg contents in soil. The red, green and gray lines with arrows represent significant positive, significant negative and nonsignificant correlations, respectively. Numbers labeled adjacent to the arrow represent the path coefficients. R^2^ is the amount of interpretation of the variable. Significance is presented by * *p* < 0.05, ** *p* < 0.01, and *** *p* < 0.001.

**Table 1 ijerph-20-03761-t001:** Changes in pH and DOC of soil amended with biochar at 300 °C, 500 °C and 700 °C pyrolysis temperatures.

Incubation Time (d)	Soil pH				DOC (mg kg^−1^)			
	CK	3BC	5BC	7BC	CK	3BC	5BC	7BC
5	4.94 a	5.00 ab	5.50 ab	5.74 b	657.22 a	691.47 b	797.47 c	866.72 c
15	5.05 a	5.76 ab	5.94 ab	6.19 b	610.63 a	913.17 bc	894.97 b	960.07 c
30	5.09 a	5.39 ab	5.86 ab	6.15 b	706.88 a	867.18 b	949.78 b	1073.68 c
45	5.62 a	5.71 a	6.01 ab	6.27 b	746.49 a	663.96 b	729.69 a	770.29 a
60	5.64 a	5.85 a	6.12 ab	6.3 b	806.69 a	678.87 b	720.59 b	773.09 ab

Data are presented as Mean, n = 3. Different letter in the same row means significant differences at *p* < 0.05.

## Data Availability

Not applicable.

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
