# Peer review of "Unveiling the Role of Dissolved Organic Matter on the Hg Phytoavailability in Biochar-Amended Soils"

_ijerph, 2023, doi:10.3390/ijerph20043761_

Round 1
Reviewer 1 Report
Review report: ijerph-2213219
The findings could be interesting for researchers. However, following comments should be addressed before proceeding this manuscript for further. Authors are strongly advised to correct manuscript as per following suggestions for enhancing readability and reproducibility of results.
1. Introduction: Indicate what has been known before and what not, and what is new and novel in this study. Include very clear study objectives and a testing hypothesis.
2. The following references maybe helpful for this paper and recommended to be cited.
https://doi.org/10.3389/feart.2022.1071228; https://doi.org/10.1021/acs.est.8b02213; https://doi.org/10.1007/s12517-022-09608-z
3. Section 2.2., Hypothesis and justification of the selection pyrolysis conditions (temperature treatments), rate of biochar and Hg concentration are not explicitly mentioned in the material and methods.
4. Lines 80-81, How have you done it? Details are missing!
5. How much soil was used? How did authors mix biochar in moisten soil?
6. How the soil properties were determined? Could you specify the methods with references added in the suplementary?
7. Please calculate the rate of biochar application per hectare and show it in the paper. Is it economical to use this amount? Please discuss about it in the main text.
8. In conclusion, add a statement for future direction.
9. In general, legends of figures and tables are not all self-explainable. I am recommending that figures must be self-explanatory. That is, all statistics and abbreviations used must be clearly explained.
Reviewer 2 Report
I rate the paper very highly because it is a simple, well-conducted experiment. It solves the serious problem of soil remediation. The topic is topical and applicable on a global scale. Although the paper is written very precisely I recommend adding the following to the methodology:
- nylon bags were inserted into the soil - the soil was where? in containers? in what size, where were they stored? Was irrigation controlled? How was soil moisture maintained at 70%?
- What am I to imagine by destructive sampling? of what? soil?
Overall, this section requires a more detailed description. The methodology should always be described to the extent that it can be reproduced.
After completion, I recommend publication.
Round 2
Reviewer 1 Report
Dear editor,
Thanks for giving m e the opportunity to re-evaluate this paper. Taking into account the reviewers comments and the revision have been done I can confirm that this manuscript has gained in value in my opinion is ready to be accepted. I would like to inform you that I am willingness to contribute in the evaluation of other papers related to my expertise to be submitted in this journal.